# Healthcare considerations for aging people in prisons

**Njideka Sanya**◉°*, **Jessica Gaber**°, **Alice Cavanagh**‡, **Michelle Howard**◉‡, **Fiona G. Kouyoumdjian**°

Department of Family Medicine, McMaster University, Hamilton, Ontario, Canada

◉ These authors contributed equally to this work.
‡ These authors also contributed equally to this work.
* sanyan@mcmaster.ca

## Abstract

The number of older adults aged 50 years and above who are incarcerated is increasing rapidly around the world. Providing high quality healthcare for aging incarcerated populations requires tailored services and approaches that consider their unique needs in carceral environments. However, there is limited research on healthcare from the perspective of relevant interest-holders, which challenges the development of interventions to address unmet needs and improve healthcare quality. This study aimed to understand the perspectives of interest-holders on the healthcare considerations of aging prison populations. We carried out secondary reflexive thematic analysis of data from interviews and focus groups with three categories of interest-holders: people with lived experience of incarceration (including people who were incarcerated at the time of data collection), community-based advocates and researchers (including people who work in community organizations, academics, and clinicians), and correctional healthcare providers and leaders. Data were collected from 61 participants between August 2022 and November 2023. We identified four themes: need for better data tracking and comparison, healthcare needs conflict with the carceral environment, physical constraints challenge access to care, and questioning the suitability of older adults for incarceration. This study highlights factors to consider in addressing healthcare for aging incarcerated populations, as described by various individuals with a vested interest in understanding and improving correctional health and health services. Findings can be used to inform ongoing work to improve healthcare and address unmet needs for aging people in prison.

## Introduction

The number of adults aged 50 and above who are incarcerated is increasing around the world [1]. In many countries, this is the fastest growing age group of the prison population, with a dramatic proportionate rise during the 21st century [2–5]. For

**Data availability statement:** The qualitative data is not publicly available due to privacy concerns in supplying the entire dataset. Requests to access the data may be considered on a case-by-case basis, subject to ethics and correctional authority approvals. Interested parties can contact Laura Cleghorn, Managing Director, Research, Department of Family Medicine, McMaster University – dfmresearch@mcmaster.ca. This data will be stored on a secure institutional SharePoint hosted by McMaster University and will be kept for seven years.

**Funding:** This project was funded by the Public Health Agency of Canada's Enhanced Surveillance for Chronic Disease Program (Grant number 2021-HQ-000098 to FGK) from 2020 to 2024. URL - https://www.canada.ca/public-health. The funders had no role in study design, data collection and analysis, decision to publish, or preparation of the manuscript.

**Competing interests:** The authors have declared that no competing interests exist.

example, in federal prisons in Canada, the proportion of people in prison aged 50 years and older nearly doubled in the last two decades, from 13% in 2000 to 25% in 2019 [6].

Providing high quality healthcare for this population requires tailored services and approaches that consider the unique physical, mental, and social needs of older adults in the context of the carceral environment [7]. For people who are aging in carceral institutions, incarceration adds a layer of complexity to the dual burdens of experiencing symptoms of a disease condition and accessing healthcare. This complexity has been identified in various age-associated health challenges including chronic conditions, [8] mental illness, [9] dementia, [10] functional health needs, [11] and palliative care [12]. Evidence suggests that people in prison often age at a faster rate and thus experience age-related declines in health earlier than their peers in the community. This phenomenon has been described as 'accelerated aging' [11,13,14]. In addition, healthcare costs are approximately three times higher for incarcerated older adults than incarcerated younger adults [15] due to a larger burden of chronic illnesses, disability, and special needs [16].

The perspectives of people who are impacted by carceral healthcare systems are valuable to inform healthcare provision and research for the aging prison population, identify priority areas for action, and significantly shape quality improvement work [17–19]. The term "interest-holders" describes "groups with legitimate interests in the health issue under consideration" [20]. In our context, this includes different types of knowledge users, collaborators, partners in Canadian correctional health, and patients (i.e., people with lived experience of incarceration).

To date, few studies have examined the opinions of people who utilize, provide, or advocate for health services for older incarcerated adults in Canada. In this paper, we aim to describe perspectives from interest-holders who discussed aging when asked about the most important health issues facing people in Canadian prisons.

## Methods

### Study design

We carried out a secondary reflexive thematic analysis [21,22] of qualitative data initially collected to identify the priorities of varied interest-holders for health surveillance of people incarcerated in Canadian prisons, as a component of a project to strengthen prison health surveillance in prisons in Canada [23,24]. This project was an integrated knowledge translation project which engaged knowledge users as active participants. Their input based on lived experience and content expertise was included in project development and especially in the design and implementation of the study [25,26]. Knowledge users contributed to the development of interview questions, identification of effective communication and recruitment strategies, and provided advice on navigating compensation for participants, especially people who were currently incarcerated. This is described in further depth in the protocol paper [23].

## Participants and setting

This study was carried out by researchers in an academic family medicine department of a university, in partnership with a Canadian correctional authority, involving collaboration and engagement with other interest-holders. We set up a Project Advisory Committee comprised of people with lived experience of incarceration and people who work in community organizations, and a Scientific/Academic team made up of people with academic and research expertise in topics related to prison health. People from these collaborating groups were first invited as participants, and subsequent recruitment was done through snowball sampling. Some collaborators shared information about the study with people they knew and asked them to reach out to us on their own or to give consent to be contacted by us, or they provided us with email addresses of people we could invite directly. People with lived and living experience of incarceration were only invited through members of the Project Advisory Committee or other participants who were community advocates or people with experience of incarceration (not by correctional staff). Some of the participants who were currently incarcerated saw the study advertised on information boards in their facilities and reached out to us on their own. Correctional staff were invited separately through an email invitation shared within their organization for interested people to contact the research team directly and they were encouraged to share the invitation to participate with other correctional staff. Potential participants were informed that participation was voluntary, and those who were not currently incarcerated were given the option of participating through an individual interview or a focus group of people in the same category.

We identified three categories of interest-holders: people with lived experience of incarceration (including those who were incarcerated at the time of data collection), community-based advocates and researchers (including people who work in community organizations, academics, and clinicians in the field), and correctional healthcare providers and management. The study includes participants from across Canada.

## Data collection

We conducted interviews and focus groups between August 16, 2022, and November 24, 2023, using a semi-structured interview guide. We received ethics approval from the Hamilton Integrated Research Ethics Board (project #14099), and additional approval from the correctional authority to invite their employees and people who were currently incarcerated as participants. All participants provided informed consent to participate prior to enrollment in the study which was either written or audio recorded. Informed consent was audio recorded for some of the participants who were currently incarcerated as per ethics board approval and due to challenges with mail and internet in the prison setting. The information and consent form was reviewed over audio recorded phone calls with currently incarcerated participants, which we had been granted a waiver to do by the ethics board. Participants were informed of the overall goals of the project only, aging and incarceration were not specifically mentioned. Demographic information was collected via REDCap for participants in the community and over the phone for currently incarcerated participants. McMaster University's REDCap servers are securely maintained in an on-site, restricted-access data center. All web-based data transfers are encrypted, and all information is stored within a private, firewall-protected network. Each user is assigned a unique login ID and password, with access privileges limited according to their designated role. Data validation methods like closed questions, required fields, and specified field ranges were applied.

Community-based participants chose either an individual interview or a focus group with other people in the same or a similar interest-holder category through Zoom or Microsoft Teams. Given challenges with internet access while in custody, people who were currently incarcerated all had individual interviews over the phone. Participants who were not affiliated with the prison authority (i.e., not employees or currently incarcerated) received a $45 gift card as a token of appreciation. Although we were not able to provide people who were currently incarcerated this honorarium due to policy, we offered each individual a certificate of appreciation as well as the option for us to donate $45 to a community-based non-profit organization of their choice or the Inmate Committee at their institution based on their preference. This strategy was suggested by project advisors including those with lived experience of incarceration. There was one request for donations to

an Inmate Committee and three to community-based advocacy groups providing legal support and education in prisons. We mailed certificates to all currently incarcerated participants.

Interviews were facilitated by one or two team members (AC, JG, SQ, NS, OV) and were 30−40 minutes long, while all focus groups were facilitated by two people and lasted for 1–1.5 hours. Field notes were made following interviews and focus groups. All phone interviews were audio recorded, while interviews and focus group sessions on Zoom and Microsoft Teams were audio and video recorded. Recordings were sent to an external transcriptionist bound by a privacy and confidentiality agreement with the university, and all transcripts were uploaded into NVivo 14 for analysis.

## Data analysis

The data from the larger study were analyzed using Braun and Clarke's six phases of reflexive thematic analysis [21,22]. Five coders (AC, JG, SQ, NS, and OV) (1) read the transcripts to familiarize themselves with the dataset; these coders were embedded in the data collection because they had also conducted interviews and facilitated focus groups; and (2) generated initial codes which were grounded in the data; each transcript was coded independently and inductively (i.e., no coding framework or codebook was developed a priori, and each transcript was coded by two coders). After we had an initial, coded, qualitative dataset, it was evident that the concept of aging in prison was a large enough theme to merit its own analysis. Two coders (JG and NS) then met to review codes and (3) generate initial themes focused on this sub-analysis of aging, using their own perspectives to develop richer insights. We then iteratively (4) developed and further reviewed themes; (5) refined, defined, and named the themes; and (6) wrote up the results with FK, with all authors contributing to the review and critique of the themes and the manuscript after that point. We applied data source triangulation (including different categories of participants to get diverse views) and investigator triangulation (two coders with different background experiences). Throughout this process all team members openly shared their perspectives and the reasoning behind selected themes. Conflicts were resolved through discussion and comparing interpretations with the original dataset. We continued to engage in iterative dialogue until we reached a consensus about the final themes.

## Rigour

We applied a number of techniques to foster rigour [27,28]. We ensured that there were participants who represented each group of interest-holders by deliberately recruiting people from each of our identified categories [24]. Open-ended questioning and sufficient time were provided during interviews for participants to reflect on and share their perspectives, and the six-step reflexive thematic analysis process was meticulously followed in analysis of the data, including exploring significant themes that were described by participants, as is the case in this paper. We engaged various interest-holders to guide our development of methods to enable participation in remote interviews by those who were currently incarcerated. Study packages which included information on specific times when research staff would be available for phone calls were mailed to potential participants, a toll-free 1–800 number was set up and added to the approved common access calling lists for each prison, we secured ethical approval for recorded verbal consent, and we explicitly acknowledged that we could not guarantee privacy or confidentiality as prison staff could monitor mail and phone communications. These strategies are outlined in more detail in a separate study publication [29]. We continued the interviews and focus groups until we were not able to recruit any more participants. By this time project members agreed that we had reached sufficient data depth, richness, and diversity [30].

## Results

### Participants

There were 61 individuals who participated in this study: 24 in individual interviews and 37 people in 11 focus groups. Participants' characteristics are described in Table 1 [24]. The age range was from 20 to 69, with most participants aged between 40–59 (67.8%). Most participants (67.9%) were women, and 75.4% were white. The majority of participants were located in

**Table 1. Participants in Interviews and Focus Groups (N=61).**

| | Participants | N (% of valid responses) | Lived experience of incarceration n (% of participants with this characteristic) |
|---|---|---|---|
| **Age[a]** | 20-29 | 6 (10.7) | 1 (16.7) |
| | 30-39 | 10 (17.9) | 1 (10) |
| | 40-49 | 19 (33.9) | 9 (47.4) |
| | 50-59 | 19 (33.9) | 10 (52.6) |
| | 60-69 | 2 (3.6) | 1(50.0) |
| | Missing | 5 | |
| **Gender** | Man | 16 (28.1) | 13 (81.2) |
| | Woman | 37 (64.9) | 8 (21.6) |
| | Non-binary | 1 (1.8) | 1 (100.0) |
| | Other | 1 (1.8) | 1 (100.0) |
| | Prefer not to respond | 2 (3.5) | 1 (50.0) |
| | Missing | 4 | |
| **Sex assigned at birth[b]** | Male | 16 (28.6) | 13 (81.2) |
| | Female | 38 (67.9) | 10 (26.3) |
| | Prefer not to respond | 2 (3.6) | 1 (50.0) |
| | Missing | 5 | |
| **Race/Ethnicity[c]** | Arab | 1 (1.8) | 0 (0.0) |
| | Black | 6 (10.5) | 5 (83.3) |
| | Indigenous | 11 (19.3) | 7 (63.6) |
| | Latin American | 1 (1.8) | 0 (0.0) |
| | West Asian | 2 (3.5) | 0 (0.0) |
| | White | 43 (75.4) | 15 (34.9) |
| | Prefer not to respond | 2 (3.5) | 1 (50.0) |
| | Missing | 4 | |
| **Current Province/Territory** | Alberta | 1 (1.8) | 0 (0.0) |
| | British Columbia | 20 (35.1) | 11 (55.0) |
| | New Brunswick | 3 (5.3) | 1 (33.3) |
| | Nova Scotia | 8 (14.0) | 4 (50.0) |
| | Ontario | 22 (38.6) | 5 (22.7) |
| | Quebec | 1 (1.8) | 1 (100.0) |
| | Saskatchewan | 2 (3.5) | 1 (50.0) |
| | Missing | 4 | |
| **Interest-holder category** | People with lived experience of incarceration | 25 (41.0) total | 25 (100.0) |
| | Currently incarcerated | *8 (13.1)* | |
| | Previously incarcerated | *17 (27.9)* | |
| | Community-based advocates and researchers | 21 (34.4) total | 5 (23.8) |
| | Academics/Clinicians | *15 (24.6)* | |
| | Community organization workers | *5 (8.2)* | |
| | Other governmental workers | *1 (1.6)* | |
| | Correctional providers | 15 (24.6) | 0 (0.0) |

[a]Only categories in which there are data are included in this table

[b]Two participants identified as non-cisgender

[c]As participants could report more than one race/ethnicity, the sum of the percentages is greater than 100%

Ontario (38.6%) or British Columbia (35.1%). Overall, 41.0% of participants had previous experience of incarceration, including 13.1% who were currently incarcerated. Correctional health care providers accounted for 24.6% of the sample, academic researchers/clinicians with expertise and experience in the field of prison health made up 24.6%, and 8.2% were people who work in community organizations that provide services and advocate for people who experience incarceration.

## Themes

We identified four themes that describe interest-holders' perspectives on healthcare considerations for the aging prison population.

### Need for better data tracking and comparison

Participants highlighted the importance of tracking conditions that are associated with aging.

"I think it's really important, given that we do have an aging population… keeping track of age-associated chronic diseases or acute diseases that are associated with aging…" (ID 8, Academic, Focus Group)

They indicated that this data would help support service planning, especially with a rapidly aging population.

"We are going to have to… implement more stuff around aging offenders and dementia… keeping tabs on cognitive functioning and… living independently… I do think the writing's on the wall… it will be [a big issue] if we don't have more of a plan" (ID 55, Correctional healthcare provider, Interview)

They also identified the importance of comparing health and disease trends in incarcerated older adults with trends for people in the community.

"The diseases associated with aging that we would see in the population who is not incarcerated that exists in the population who is incarcerated as well, and probably to a much greater extent." (ID 14, Clinician, Interview)

### Physical constraints challenge access to care

Participants described accessing care while incarcerated as more challenging for older adults who are not always able to cope with the physical demands of getting to the location of the healthcare provider.

"…a 65-year-old guy on crutches… in the unit that is the furthest away from healthcare in the entire place… he's got to walk two football fields to get to healthcare. And he's on medications, lots of them… he's a cancer survivor. [He has to] hobble up to healthcare on crutches first thing in the morning…" (ID 51, Person who is incarcerated, Interview)

Physical constraints could prevent people from seeking or accessing care when they need it, which is especially meaningful in a carceral environment where line-ups are common.

"The older folk have a harder time trying to get what they're looking for because the line-ups are so long… that they just end up giving up, and just walking away." (ID 49, Person who is incarcerated, Interview)

### Healthcare needs conflict with carceral environment

Participants highlighted the need for specialized fittings and mobility devices associated with older age as conflicting with the restrictions of the prison environment.

"We have offenders with life sentences… retirement age… getting old… they will begin to have just some of the problems that we have as we get old: mobility concerns, fall-risk, additional needs for, you know, just even grasping bars, or reach-grabbers, different tools." (ID 47, Correctional healthcare manager, Interview)

They also described the need for routine preventive healthcare services to be made available for people aging in prison.

"…what are the protocols? … [for] the individual that reaches 50, what is the work-up that's being done for background medical conditions…?" (ID 44, Community based advocate, interview)

Additional needs described reflect specific conditions in custody, such as the quality of mattresses.

"Aging… it's worse and worse… without the proper supports… something as simple as the mattresses are… thin… [One woman] was in a cell by herself, [she] took the other mattress and put it on her bed… and she got in trouble for that." (ID 4, Community based advocate, Focus Group)

### Questioning the suitability of older adults for incarceration

Overall, participants questioned the suitability of older adults for incarceration at all, especially for people with health conditions like dementia or physical disabilities.

"Is it appropriate to incarcerate individuals that have a very high need and a complex mental health issue [dementia]? To be inside in the first place?" (ID 44, Community based advocate, Interview)

They questioned keeping high numbers of aging adults in prison, questioning the need for incarceration based on perceived risk of further criminal justice system involvement.

"The population is quickly turning into senior citizens as the largest demographic of prisoners, which are probably among the lowest risk. So, it doesn't really make a lot of sense." (ID 33, Person with lived experience of incarceration, Focus Group)

Participants also questioned the feasibility of meeting the medical needs of aging adults in the environment of correctional facilities:

"…this dance we have to do between accommodating their medical concerns as an aging person, while also being very cognizant of the security-mindedness of our environment, does make a little bit of a challenge…" (ID 47, Correctional healthcare manager, Interview)

## Discussion

In this study, we explored the perspectives of interest-holders who identified the need for greater focus on older adults in response to being asked about the most important health issues in Canadian prisons as part of a larger study. Participants described the need to improve data tracking, the challenges with accessing care due to physical constraints, how aging-specific healthcare needs conflict with the prison environment, and they questioned the suitability of older adults for incarceration at all.

Our findings align with data from other studies and reports on the incarcerated aging population. The 2019 report of the Office of the Correctional Investigator in Canada described the lack of tracking healthcare costs by age, challenges with accessibility, long distances to healthcare services, staff who are untrained in age-related health issues like dementia, and facilities which were not built to house older persons [6]. Other studies have highlighted chronic disease management, mental health, and geriatric syndromes like vision and hearing loss, cognitive impairment, urinary incontinence, and falls as challenges faced by aging people in custody [16,31].

The need for systemic data collection on the health status of incarcerated people who are aging has been identified by various authors [15,17,32]. Participants in our study wanted improvements in the collection and tracking of data on the (increasing) size of the aging population and age-specific health conditions, to ensure that carceral institutions understood their health needs and plan adequately to meet health and healthcare needs. Ahalt et al. proposed that collecting data on healthcare costs and outcomes according to chronic medical condition, healthcare delivery site, and sentence in incarcerated people aged 50 and above would enable evaluation and improvement of geriatric prison health by clinicians and policy makers [15].

The dissonance between providing adequate care for older adults while satisfying the restrictive requirements of the prison environment, articulated by one participant as "this dance we have to do…," has been described elsewhere. Common age-related health problems with mobility may be exacerbated by conditions of confinement like shackling for transport, which increases the risk of falls [7,16]. Incarcerated older adults may not have sufficient support or adequate time to complete activities of daily living (bathing, toileting, eating) [16,33], and further have to contend with unique prison-specific activities, described as 'prison activities of daily living,' [11] like standing for head count, dropping to the floor for alarms, hearing orders from staff, and climbing on top bunks. People with dementia may unintentionally engage in prohibited behaviours, which may result in punitive actions against them [34]. There have been recommendations for the consideration of release options for older and long-serving offenders who do not pose undue risks to public safety [6], supported by data showing low rates of recidivism in this population [35], the rising costs of incarcerating older adults, and the need to ensure alignment with international human rights guidelines, such as the obligation in the Nelson Mandela rules to practice non-discrimination by taking individual needs into account, especially for the most vulnerable [36,37].

The data in this study were derived from a large qualitative study with interest-holders from varied groups. We note that although this paper presents a secondary data analysis, there was an apparent convergence of opinions regarding health and healthcare needs for incarcerated people who were aging in prison, across interest-holder groups. One study limitation is that it does not include a large number of older adults with lived experience of incarceration and representatives of organizations focused on aging; this is a sub analysis of data collected for a study that did not specifically invite people from these groups. Also, the use of snowball sampling may have led to selection bias in this study [38], which could limit how well the participants reflect the broader incarcerated population. Another limitation was that the participants who were currently incarcerated were all in men's prisons, though we did include people with previous experience of incarceration in women's prisons.

Interventions like specialized facilities have been introduced in some centres to provide appropriate healthcare for incarcerated older adults, but in Canada these facilities had challenges with staff availability, selection, training, and the absence of appropriate policy or oversight [6,39]. In addition, there is limited research that could inform the improvement of interventions like this that aim to address the healthcare needs of the aging prison population [40].

## Conclusion

This work contributes to the growing body of research on the healthcare challenges for aging prison populations in Canada and around the world. Our findings demonstrate that there is a considerable amount of concern about aging while incarcerated among correctional healthcare interest-holders in correctional healthcare in Canada, and highlights factors that need to be addressed. This study can be used to inform ongoing policy reforms by policymakers, academics,

clinicians, and older adults with lived experience of incarceration. While this paper describes interest-holder perspectives on healthcare needs, it does not explore their perspectives on existing or potential options to address these needs. Additional research on potential interventions to address unmet needs would be valuable, and in particular should engage and include older people with lived and living experience of incarceration.

## Supporting information

**S1 Text. Published protocol paper.**
(PDF)

**S2 Text. Published primary study.**
(PDF)

## Acknowledgments

We gratefully acknowledge all people who contributed to the development, planning, and implementation of this study including members of the Project Advisory Council and Project Implementation Team. We also acknowledge Olivia Virag and Seth Walvinz Quimson who supported the qualitative analysis. We also very gratefully acknowledge the participants who navigated the challenges of the prison context alongside us in order to take part in this study.

## Author contributions

**Conceptualization:** Njideka Sanya, Jessica Gaber, Fiona G. Kouyoumdjian.

**Data curation:** Njideka Sanya, Jessica Gaber, Alice Cavanagh.

**Formal analysis:** Njideka Sanya, Jessica Gaber, Alice Cavanagh, Michelle Howard, Fiona G. Kouyoumdjian.

**Funding acquisition:** Fiona G. Kouyoumdjian.

**Methodology:** Jessica Gaber, Fiona G. Kouyoumdjian.

**Writing – original draft:** Njideka Sanya, Jessica Gaber, Fiona G. Kouyoumdjian.

**Writing – review & editing:** Njideka Sanya, Jessica Gaber, Alice Cavanagh, Michelle Howard, Fiona G. Kouyoumdjian.

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
