## [Decision Letter · Decision Letter 0]

19 Jan 2026

PONE-D-25-48938Healthcare considerations for aging people in prisonsPLOS One

Dear Dr. Sanya,

Thank you for submitting your manuscript to PLOS ONE. After careful consideration, we feel that it has merit but does not fully meet PLOS ONE’s publication criteria as it currently stands. Therefore, we invite you to submit a revised version of the manuscript that addresses the points raised during the review process

**Reviewer comments are set out at the bottom of this email.**

A letter that responds to each point raised by the reviewer(s). You should upload this letter as a separate file labeled 'Response to Reviewers'.A marked-up copy of your manuscript that highlights changes made to the original version. You should upload this as a separate file labeled 'Revised Manuscript with Track Changes'.An unmarked version of your revised paper without tracked changes. You should upload this as a separate file labeled 'Manuscript'.

We look forward to receiving your revised manuscript.

Kind regards,

Steph Scott

Academic Editor

PLOS One

**Journal Requirements:**

1. When submitting your revision, we need you to address these additional requirements. Please ensure that your manuscript meets PLOS ONE's style requirements, including those for file naming. The PLOS ONE style templates can be found at https://journals.plos.org/plosone/s/file?id=wjVg/PLOSOne_formatting_sample_main_body.pdf and https://journals.plos.org/plosone/s/file?id=ba62/PLOSOne_formatting_sample_title_authors_affiliations.pdf 2. We noticed you have some minor occurrence of overlapping text with the following previous publication(s), which needs to be addressed:https://www.sciencedirect.com/science/article/pii/S0033350625003439?via%3DihubIn your revision ensure you cite all your sources (including your own works), and quote or rephrase any duplicated text outside the methods section. Further consideration is dependent on these concerns being addressed. 3. We noted in your submission details that a portion of your manuscript may have been presented or published elsewhere. “The participants' characteristics described in Table 1 (Participants in Interviews and Focus Groups (N=61)) are included in a published manuscript that reports the results from the primary study. This does not constitute dual publication because both manuscripts only contain a similar description of the participants' demographic characteristics as supporting information to appropriately interpret the complete results and analysis”.  Please clarify whether this [conference proceeding or publication] was peer-reviewed and formally published. If this work was previously peer-reviewed and published, in the cover letter please provide the reason that this work does not constitute dual publication and should be included in the current manuscript. 4. We note that you have indicated that there are restrictions to data sharing for this study. For studies involving human research participant data or other sensitive data, we encourage authors to share de-identified or anonymized data. However, when data cannot be publicly shared for ethical reasons, we allow authors to make their data sets available upon request. For information on unacceptable data access restrictions, please see http://journals.plos.org/plosone/s/data-availability#loc-unacceptable-data-access-restrictions. Before we proceed with your manuscript, please address the following prompts: a) If there are ethical or legal restrictions on sharing a de-identified data set, please explain them in detail (e.g., data contain potentially identifying or sensitive patient information, data are owned by a third-party organization, etc.) and who has imposed them (e.g., a Research Ethics Committee or Institutional Review Board, etc.). Please also provide contact information for a data access committee, ethics committee, or other institutional body to which data requests may be sent. b) If there are no restrictions, please upload the minimal anonymized data set necessary to replicate your study findings to a stable, public repository and provide us with the relevant URLs, DOIs, or accession numbers. Please see http://www.bmj.com/content/340/bmj.c181.long for guidelines on how to de-identify and prepare clinical data for publication. For a list of recommended repositories, please see https://journals.plos.org/plosone/s/recommended-repositories. You also have the option of uploading the data as Supporting Information files, but we would recommend depositing data directly to a data repository if possible. Please update your Data Availability statement in the submission form accordingly. 5. Please amend either the abstract on the online submission form (via Edit Submission) or the abstract in the manuscript so that they are identical. 6. We note that there is identifying data in Table 1. Due to the inclusion of these potentially identifying data, we have removed this file from your file inventory. Prior to sharing human research participant data, authors should consult with an ethics committee to ensure data are shared in accordance with participant consent and all applicable local laws. Data sharing should never compromise participant privacy. It is therefore not appropriate to publicly share personally identifiable data on human research participants. The following are examples of data that should not be shared: -Name, initials, physical address-Ages more specific than whole numbers-Internet protocol (IP) address-Specific dates (birth dates, death dates, examination dates, etc.)-Contact information such as phone number or email address-Location data-ID numbers that seem specific (long numbers, include initials, titled “Hospital ID”) rather than random (small numbers in numerical order) Data that are not directly identifying may also be inappropriate to share, as in combination they can become identifying. For example, data collected from a small group of participants, vulnerable populations, or private groups should not be shared if they involve indirect identifiers (such as sex, ethnicity, location, etc.) that may risk the identification of study participants. Additional guidance on preparing raw data for publication can be found in our Data Policy (https://journals.plos.org/plosone/s/data-availability#loc-human-research-participant-data-and-other-sensitive-data) and in the following article: http://www.bmj.com/content/340/bmj.c181.long. Please remove or anonymize all personal information (<specific identifying information in file to be removed>), ensure that the data shared are in accordance with participant consent, and re-upload a fully anonymized data set. Please note that spreadsheet columns with personal information must be removed and not hidden as all hidden columns will appear in the published file. 7. Please include captions for your Supporting Information files at the end of your manuscript, and update any in-text citations to match accordingly. Please see our Supporting Information guidelines for more information: http://journals.plos.org/plosone/s/supporting-information. 8. If the reviewer comments include a recommendation to cite specific previously published works, please review and evaluate these publications to determine whether they are relevant and should be cited. There is no requirement to cite these works unless the editor has indicated otherwise.

Reviewers' comments:

Reviewer's Responses to Questions

**Comments to the Author**

1. Is the manuscript technically sound, and do the data support the conclusions?

Reviewer #1: Yes

Reviewer #2: Partly

2. Has the statistical analysis been performed appropriately and rigorously?

Reviewer #1: N/A

Reviewer #2: N/A

3. Have the authors made all data underlying the findings in their manuscript fully available?

Reviewer #1: No

Reviewer #2: No

4. Is the manuscript presented in an intelligible fashion and written in standard English?

Reviewer #1: Yes

Reviewer #2: Yes

5. Review Comments to the Author

**Reviewer #1:** Thank you for the opportunity to review this paper reporting the results of secondary data analysis to explore perspectives of healthcare considerations for aging prisoners in Canada

Overall this is a well presented report with clear aims, methods, results, implications and limitations. The authors are appropriate in the modest conclusions drawn.

Specific comments

- could the abstract make the case for why this study matters, it doesn't currently strike me as an important study to have done, although the manuscript itself unpacks some of this

- in the 'rigour' section you state you ensured participants were representative of the various interest holders, how did you do this?

- relatedly, in Table 1, could you present demographics by interest group, or at least separating those with imprisonment experience so that representativeness can be assessed. You say this is representative, but (for example) 65% female is not representative of the prison experienced population.

**Reviewer #2:** The authors state that the qualitative data are not publicly available due to ethical restrictions and participant privacy, which is understandable. To support transparency and FAIR principles, it would be helpful to clarify if controlled access for qualified researchers is possible, or if anonymized summaries or coded datasets could be shared to facilitate reproducibility.

The authors state that this was (part of) an integrated knowledge translation (IKT) project. Could the authors clarify how knowledge users contributed to study design; for example, in formulating questions, developing instruments, choosing methods, or advising on procedures?

A roject Advisory Committee and a Scientific/Academic team were established, and the researchers used snowball sampling to recruit participants across Canada. While this approach facilitates stakeholder engagement and access to a hard-to-reach population, it raises some methodological and ethical considerations. Snowball sampling may introduce selection bias and limit representativeness. It is unclear to which extent this approach also created potential role conflicts. Clarification of measures to protect vulnerable participants and ensure voluntary participation would also be valuable.

Could the authors provide more details on how REDCap was used, including instrument setup, data validation features, and measures to ensure security and participant confidentiality?

The approach to participant recognition is thoughtful and ethically sound. From a methodological perspective, it would be interesting to know how participants responded to the certificates and donation option, and whether this influenced engagement, openness, or the richness of the data collected. Any observations or reflections on participants’ reactions could provide valuable insight into the impact of these ethical and practical choices on the study process.

Could the authors further clarify the process used to develop the themes, e.g., how consensus was reached and steps taken to reduce bias? Additionally, could the authors explain why knowledge users were not involved in data analysis or interpretation, as their involvement could enhance contextual understanding and the actionable relevance of findings in line with Integrated Knowledge Translation principles?

The authors describe rigorous data collection and reflexive thematic analysis. Could they clarify how representativeness of participants was ensured given the snowball sampling approach? Additionally, please specify whether any steps were taken to address limitations of remote interviews. Further detail on any formal criteria for saturation or participant validation procedures would strengthen transparency and methodological rigor.

The results identify four themes supported by illustrative quotes from multiple stakeholder groups, which is valuable. However, it is not fully clear how these themes were developed from the data. The manuscript provides limited detail on the coding process, how consensus was reached among researchers, and whether any form of triangulation or validation (e.g., member checking, peer debriefing) was applied. It is also unclear how representative the quoted examples are of the broader participant sample or whether minority or conflicting perspectives were considered. Without this information, it is difficult to fully assess the rigor and robustness of the findings. Providing a more detailed description of the analytic procedures and how data support the themes would strengthen confidence in the results.

The results section identifies only four themes, yet each theme is illustrated by only two quotes and there is very little interpretative analysis. This limited presentation makes it difficult to evaluate whether the findings are robust or representative of the full dataset. Without richer data support, exploration of conflicting perspectives, or deeper analytical discussion, the results appear largely descriptive and provide minimal insight beyond what is already known. The manuscript would be strengthened by including more illustrative quotes, detailed explanation of theme development, discussion of (possibly divergent) perspectives, and interpretative analysis linking findings to broader literature or theoretical frameworks.

Hte study presents a secondary analysis of an existing dataset. Could the authors clarify how the original data were suited to address the current research questions on aging in prison populations? It would be helpful to know what steps were taken to maintain rigor and validity, and how limitations of the original dataset (such as potential gaps in perspectives or data not originally collected for this focus) were addressed. Providing this information would strengthen confidence in the robustness and interpretability of the findings

6. PLOS authors have the option to publish the peer review history of their article (what does this mean?). If published, this will include your full peer review and any attached files.

Reviewer #1: **Yes:** DR CATRIONA CONNELL

Reviewer #2: No

---

## [Author Response · Author response to Decision Letter 1]

19 Mar 2026

Dear reviewers,

We appreciate the helpful comments from the reviewers and editors, and we have made revisions to address the comments. Please see below for itemized responses to the comments and questions.

REVIEWER 1

Could the abstract make the case for why this study matters, it doesn't currently strike me as an important study to have done, although the manuscript itself unpacks some of this

We have edited the abstract to include this: “However, there is limited research on healthcare from the perspective of relevant interest-holders, which challenges the development of interventions to address unmet needs and improve healthcare quality.”

In the 'rigour' section you state you ensured participants were representative of the various interest holders, how did you do this?

For clarity, we have taken out the clause “participants were representative of the variety of interest-holders” and rewritten the sentence as “We ensured that there were participants who represented each group of interest-holders by deliberately recruiting people from each of our identified categories”, which better describes our intended meaning. We also reworded other sections of the paper where the word ‘representative’ was used. We also acknowledged the risks associated with our participant recruitment strategy as a potential limitation in the Discussion.

- relatedly, in Table 1, could you present demographics by interest group, or at least separating those with imprisonment experience so that representativeness can be assessed. You say this is representative, but (for example) 65% female is not representative of the prison experienced population.

We have edited Table 1 to separate those with lived experience of incarceration. Also, we have removed our use of the term ‘representative’ as described above.

REVIEWER 2

The authors state that the qualitative data are not publicly available due to ethical restrictions and participant privacy, which is understandable. To support transparency and FAIR principles, it would be helpful to clarify if controlled access for qualified researchers is possible, or if anonymized summaries or coded datasets could be shared to facilitate reproducibility.

The qualitative data is not publicly available due to privacy concerns in supplying the entire dataset. Requests to access the data may be considered on a case-by-case basis, subject to ethics and correctional authority approvals. Interested parties can contact Laura Cleghorn, Managing Director, Research, Department of Family Medicine, McMaster University – dfmresearch@mcmaster.ca. This data will be stored on a secure institutional SharePoint hosted by McMaster University and will be kept for seven years.

The authors state that this was (part of) an integrated knowledge translation (IKT) project. Could the authors clarify how knowledge users contributed to study design; for example, in formulating questions, developing instruments, choosing methods, or advising on procedures?

For clarity, we have included this explanation that “Knowledge users contributed to the development of interview questions, identification of effective communication and recruitment strategies, and provided advice on navigating compensation for participants, especially people who were currently incarcerated. This is described in further depth in the protocol paper.”

A Project Advisory Committee and a Scientific/Academic team were established, and the researchers used snowball sampling to recruit participants across Canada. While this approach facilitates stakeholder engagement and access to a hard-to-reach population, it raises some methodological and ethical considerations. Snowball sampling may introduce selection bias and limit representativeness. It is unclear to which extent this approach also created potential role conflicts. Clarification of measures to protect vulnerable participants and ensure voluntary participation would also be valuable.

We have included more details about the participant recruitment strategy: “Some collaborators shared information about the study with people they knew and asked them to reach out to us on their own or to give consent to be contacted by us, or they provided us with email addresses of people we could invite directly. People with lived and living experience of incarceration were only invited through members of the Project Advisory Committee or other participants who were community advocates or people with experience of incarceration (not by correctional staff). Some of the participants who were currently incarcerated saw the study advertised on information boards in their facilities and reached out to us on their own. Correctional staff were invited separately through an email invitation shared within their organization for interested people to contact the research team directly and were encouraged to share the invitation to participate with other correctional staff. Potential participants were informed that participation was voluntary, and those who were not currently incarcerated were given the option of participating through an individual interview or a focus group of people in the same category.

We also added an acknowledgement of the potential selection bias due to snowball sampling as a limitation in the discussion section: “Also, the use of snowball sampling may have led to selection bias in this study, which could limit how well the participants reflect the broader population.

Could the authors provide more details on how REDCap was used, including instrument setup, data validation features, and measures to ensure security and participant confidentiality?

We have added more details with the following sentences: “McMaster University’s REDCap servers are securely maintained in an on‑site, restricted‑access data center. All web‑based data transfers are encrypted, and all information is stored within a private, firewall‑protected network. Each user is assigned a unique login ID and password, with access privileges limited according to their designated role. Data validation methods like closed questions, required fields, and specified field ranges were applied.”

The approach to participant recognition is thoughtful and ethically sound. From a methodological perspective, it would be interesting to know how participants responded to the certificates and donation option, and whether this influenced engagement, openness, or the richness of the data collected. Any observations or reflections on participants’ reactions could provide valuable insight into the impact of these ethical and practical choices on the study process.

Thank you for this comment. We have provided details of the responses to the participant recognition options thus: “This strategy was suggested by project advisors including those with lived experience of incarceration. There was one request for donations to an Inmate Committee and three to community-based advocacy groups providing legal support and education in prisons. We mailed certificates to all currently incarcerated participants.”

Could the authors further clarify the process used to develop the themes, e.g., how consensus was reached and steps taken to reduce bias? Additionally, could the authors explain why knowledge users were not involved in data analysis or interpretation, as their involvement could enhance contextual understanding and the actionable relevance of findings in line with Integrated Knowledge Translation principles?

We have included further details of methods to improve qualitative rigour, the theme development process, and managing consensus and conflicts: “We applied data source triangulation (including different categories of participants to get diverse views) and investigator triangulation (two coders with different background experiences). Throughout this process all team members openly shared their perspectives and the reasoning behind selected themes. Conflicts were resolved through discussion and comparing interpretations with the original dataset. We continued to engage in iterative dialogue until we reached a consensus about the final themes.”

While knowledge users were involved in data interpretation of the primary study, this paper was prepared when the grant funding period was drawing to a close and the Project Advisory Committee was being dissolved. At this point we could no longer reimburse members for time spent on the project.

The authors describe rigorous data collection and reflexive thematic analysis. Could they clarify how representativeness of participants was ensured given the snowball sampling approach? Additionally, please specify whether any steps were taken to address limitations of remote interviews. Further detail on any formal criteria for saturation or participant validation procedures would strengthen transparency and methodological rigor.

We have described our clarification of the use of the term ‘representativeness of participants’ above. We have added the following statements to the rigour section: “We engaged various interest-holders to guide our development of methods to support participation in remote interviews by those who were currently incarcerated. Study packages which included information on specific times when research staff would be available for phone calls were mailed to potential participants, a toll-free 1-800 number was set up and added to the approved common access calling lists for each prison, secured ethical approval for recorded verbal consent, and explicitly acknowledged that we could not guarantee privacy or confidentiality as prison staff could monitor mail and phone communications. These strategies are outlined in more detail in this published paper. [29]”

We were not working to reach saturation as it is not a recommended metric in Braun & Clarke’s reflexive thematic analysis, but instead we used the metric: “By this time all project members agreed that we had reached sufficient data depth, richness, and diversity. [30]”

The results identify four themes supported by illustrative quotes from multiple stakeholder groups, which is valuable. However, it is not fully clear how these themes were developed from the data. The manuscript provides limited detail on the coding process, how consensus was reached among researchers, and whether any form of triangulation or validation (e.g., member checking, peer debriefing) was applied. It is also unclear how representative the quoted examples are of the broader participant sample or whether minority or conflicting perspectives were considered. Without this information, it is difficult to fully assess the rigor and robustness of the findings. Providing a more detailed description of the analytic procedures and how data support the themes would strengthen confidence in the results.

We have included further details about theme development, triangulation, and managing conflict and consensus, as described above.

The results section identifies only four themes, yet each theme is illustrated by only two quotes and there is very little interpretative analysis. This limited presentation makes it difficult to evaluate whether the findings are robust or representative of the full dataset. Without richer data support, exploration of conflicting perspectives, or deeper analytical discussion, the results appear largely descriptive and provide minimal insight beyond what is already known. The manuscript would be strengthened by including more illustrative quotes, detailed explanation of theme development, discussion of (possibly divergent) perspectives, and interpretative analysis linking findings to broader literature or theoretical frameworks.

We have added further explanation of theme development (as above), and further interpretive analysis of the findings in the text.

The study presents a secondary analysis of an existing dataset. Could the authors clarify how the original data were suited to address the current research questions on aging in prison populations? It would be helpful to know what steps were taken to maintain rigor and validity, and how limitations of the original dataset (such as potential gaps in perspectives or data not originally collected for this focus) were addressed. Providing this information would strengthen confidence in the robustness and interpretability of the findings

We have reframed the aim of this study to: “In this paper, we aim to describe perspectives from interest-holders who discussed aging when asked about the most important health issues facing people in Canadian prisons.”

We have included information about our steps to maintain rigor and validity in responses to other questions.

Editor’s comments to the author

We have reviewed and made revisions based on PLOS ONE’s style requirements for our manuscript and file naming for supporting information.

We noticed you have some minor occurrence of overlapping text with the following previous publication(s), which needs to be addressed:

https://www.sciencedirect.com/science/article/pii/S0033350625003439?via%3Dihub

In your revision ensure you cite all your sources (including your own works), and quote or rephrase any duplicated text outside the methods section. Further consideration is dependent on these concerns being addressed.

Thank you for pointing this out. We have cited this publication appropriately and rephrased duplicated text.

We noted in your submission details that a portion of your manuscript may have been presented or published elsewhere. “The participants' characteristics described in Table 1 (Participants in Interviews and Focus Groups (N=61)) are included in a published manuscript that reports the results from the primary study. This does not constitute dual publication because both manuscripts only contain a similar description of the participants' demographic characteristics as supporting information to appropriately interpret the complete results and analysis”. Please clarify whether this [conference proceeding or publication] was peer-reviewed and formally published. If this work was previously peer-reviewed and published, in the cover letter please provide the reason that this work does not constitute dual publication and should be included in the current manuscript.

The participants’ characteristics outlined in Table 1 were included in another peer-reviewed publication that describes results from the primary study which identified interest-holder priorities for health surveillance in prisons. Although both manuscripts contain a similar description of the participants’ demographic characteristics, we think it is important to report the same information in both manuscripts.

We note that you have indicated that there are restrictions to data sharing for this study. For studies involving human research participant data or other sensitive data, we encourage authors to share de-identified or anonymized data. However, when data cannot be publicly shared for ethical reasons, we allow authors to make their data sets available upon request. For information on unacceptable data access restrictions, please see http://journals.plos.org/plosone/s/data-availability#loc-unacceptable-data-access-restrictions.

b) If there are no restrictions, please upload the minimal anonymized data set necessary to replicate your study findings to a stable, public repository and prov

---

## [Decision Letter · Decision Letter 1]

8 Apr 2026

Healthcare considerations for aging people in prisons

PONE-D-25-48938R1

Dear Dr. Sanya,

We’re pleased to inform you that your manuscript has been judged scientifically suitable for publication and will be formally accepted for publication once it meets all outstanding technical requirements.

Kind regards,

Steph Scott

Academic Editor

PLOS One

Additional Editor Comments (optional):

Reviewers' comments:

Reviewer's Responses to Questions

**Comments to the Author**

1. If the authors have adequately addressed your comments raised in a previous round of review and you feel that this manuscript is now acceptable for publication, you may indicate that here to bypass the “Comments to the Author” section, enter your conflict of interest statement in the “Confidential to Editor” section, and submit your "Accept" recommendation.

Reviewer #1: All comments have been addressed

2. Is the manuscript technically sound, and do the data support the conclusions?

Reviewer #1: Yes

3. Has the statistical analysis been performed appropriately and rigorously?

Reviewer #1: N/A

4. Have the authors made all data underlying the findings in their manuscript fully available?

Reviewer #1: Yes

5. Is the manuscript presented in an intelligible fashion and written in standard English?

Reviewer #1: Yes

6. Review Comments to the Author

Reviewer #1: I am content that my peer review comments from the prior manuscript have been addressed and to recommend accepting this manuscript. There is a minor issue with repetition of ‘or’ around line 105, and repetition of ‘questioning’ in the added text.

7. PLOS authors have the option to publish the peer review history of their article (what does this mean?). If published, this will include your full peer review and any attached files.

Reviewer #1: No

---

## [Editor Report · Acceptance letter]

PONE-D-25-48938R1

PLOS One

Dear Dr. Sanya,

I'm pleased to inform you that your manuscript has been deemed suitable for publication in PLOS One. Congratulations! Your manuscript is now being handed over to our production team.

Kind regards,

on behalf of

Dr. Steph Scott

Academic Editor

PLOS One